# A Novel Approach for Selecting Effective Threshold Values in Ternary State Estimation Using Particle Swarm Optimization

Somayeh Davar  and Thomas Fevens *

Faculty of Computer Science and Software Engineering, Concordia University, Montréal, QC H3G 1M8, Canada
* Correspondence: fevens@encs.concordia.ca

**Abstract:** Inspired by recent breakthroughs in cyber-physical systems (CPSs) and their applications, in this paper, we propose a novel multi-objective method to optimize the threshold values within the ternary event-based framework. To reduce communication overhead, the particle swarm optimization (PSO) approach is applied as an optimizer to identify Pareto optimal set values of the threshold. The proposed optimization technique is subject to constraints to ensure its feasibility. The simulation results confirm the efficiency of the recommended method. Furthermore, the simulation results demonstrate that the proposed framework is comprehensive and capable of finding a wide variety of Pareto optimal ternary event-based state estimations for each predefined threshold.

**Keywords:** event-based state estimation; wireless sensor networks; ternary and hybrid event-based; multi-objective optimization; swarm optimization

## 1. Introduction

Cyber-physical systems (CPSs), and the integration of event-based estimation (EBE) methodologies in this framework, have gained considerable attraction by researchers in recent years [1,2]. In particular, the development of strategies for energy-efficient data transmission of sensor observations in wireless sensor networks (WSNs) in order to decrease the communication overhead is essential in response to the recent progress in CPSs. Some of the application areas of WSNs include biosensors for healthcare monitoring [3], remote monitoring and surveillance [4], wildfire monitoring [5], intelligent environment and Internet of Things (IoT) [6], and target tracking [7]. The communication potential, as well as the energy of the sensor nodes in WSNs, are constrained due to cost and other considerations, including storage capacity and communication bandwidth. Therefore, in WSNs, information transmission and computing power must be lowered to conserve energy. As a result, having low-energy and high-performance WSNs are necessary. Yan et al. [8], for instance, suggested a low energy-based node positioning method in WSNs, based on the PSO approach. The purpose of their suggested technique is to efficiently identify and optimize the dynamic location of the sensor nodes, which may considerably enhance routing in terms of decreasing the energy consumption of the optical sensor nodes. Additionally, an adaptive distributed artificial intelligence (ADAI) method, a hierarchical resource allocation strategy along with adaptive particle swarm optimization (APSO), was introduced by Mukherjee et al. [9] in order to solve the problem of energy consumption and resource allocation in multi-agent clustering networks. Over the past few decades, various approaches for event-based state estimates have been developed by researchers to address the sensor's data transmission efficiency issue, besides other issues of a similar nature, such as the power constraint of a remote sensor and its limited bandwidth [10–16].

In an EBE framework, the sensor's data is only transmitted to the fusion center (FC), or neighboring nodes, upon the occurrence of a specific event determined through an event-triggering mechanism. Two primary challenges need to be addressed for an EBE framework: (i) First, if the triggering event is not satisfied, resulting in the absence of

an observation transmitted to the FC, the estimator can still read a measurement resulting from an event set defined by the triggering mechanism as secondary information. Including this secondary information in the interim of a non-event, iterations lead to a strategy that updates the state estimation in a hybrid fashion, simultaneously employing the measurements as set and point values. As a result of this hybrid scenario, the posterior distribution becomes non-Gaussian. One primary strategy to overcome this problem is to employ stochastic triggering [17] for transmitting observations, which results in a Gaussian approximation [18] of the posterior distribution. (ii) Second, EBE strategies include a binary decision method on whether or not the sensor communicates its raw measurement during an epoch (defined as the time frame between consecutive sensor activations and based on the EBE strategy to decide when to transmit the observation, i.e., event epochs or idle epochs) since transferring raw data during all the event occurrences would be inefficient. As a result, the sensor communicates its raw measurement during an event epoch while retaining its local measurement during an idle epoch [13]. To address the aforementioned challenge Davar et al. [14,19] introduced a different approach, known as the ternary event-triggering (TET) approach. Likewise, instead of a stochastic triggering technique, a deterministic triggering mechanism via thresholding, referred to as send on delta (SOD) [20,21], has been utilized to convert the non-Gaussian posterior distribution to its Gaussian equivalent. The key rationale behind this choice is to provide a more accurate estimation of the event-based posterior distribution than a Gaussian estimation provided by a stochastic triggering [22]. As a result, the fact that a definitive triggering method (e.g., SOD) is dependent on a predetermined threshold becomes a critical issue. This research aims to develop a novel multi-objective framework, called ternary event triggering swarm optimization (TETSO), to improve the TET mechanism's threshold values. The objectives of this paper improve the state of the art by the following: (1) Automating the process for selecting threshold values versus with human intervention; (2) Finding the optimized threshold values for the TET mechanism; (3) Reducing the communication overhead; (4) Improving the energy efficiency for both the sensor side and the remote estimator, and (5) Requiring less maintenance and, as a consequence, a decrease in the overall cost.

The significant contribution of this new approach is the lowering of the communication overhead by optimizing threshold levels for the TET mechanism. To achieve this goal, a multi-objective approach is developed to increase the transmission rate of quantized observations while simultaneously sending minimal periodic data during idle epochs, while avoiding the transmission of redundant information, which, in turn, results in preserving limited power supplies.

The main contributions of the proposed work can be summarized as follows:

- An automated and efficient thresholding event triggering swarm optimization (TETSO) technique is proposed.
- Multi-objective PSO is utilized to improve boundary selection of the TET mechanism.
- The efficiency of the proposed TETSO method is confirmed with simulation results.

The rest of this work is structured as follows: Section 2 presents the background information and the problem formulation. The suggested TETSO architecture is introduced in Section 3. Simulation results and their interpretation with regards to performance are given in Section 4. Finally, we conclude this paper in Section 5.

## 2. Problem Formulation

In the EBE context, consider a dynamic system whose state is measured by multiple sensors. For instance, several sensors located in different locations on the ground can simultaneously observe an object in a larger space area. Each sensor can be positioned far or near the object, and the transmission of information can be fast or slow. Furthermore, once the sensor has collected its data, this data must be delivered to the fusion center for further processing.

Therefore the following open-loop state-space model is considered which used to model the estimation problem:

$$x_k = \mathbf{F}_k x_{k-1} + \mathbf{w}_k \tag{1}$$
$$z_k = \mathbf{H}_k x_k + \mathbf{v}_k \tag{2}$$

where $k$ denotes an iteration index; $x_k \in \mathbb{R}^{n_x}$ denotes the state vector; and $z_k \in \mathbb{R}^{n_z}$ represents the sensor's observations. Functions $\mathbf{H}_k$ and $\mathbf{F}_k$ show the observation and state models, respectively. Uncertainties in the observation and state models are represented by $w_k$ and $v_k$, respectively. These uncertainties are considered mutually uncorrelated white Gaussian noises having covariances $\mathcal{Q}_k > 0$ and $\mathcal{R}_k > 0$, respectively. The state-space model represented in Equations (1) and (2) is based on the development presented by Joris and Mircea [23]. An important constraint in event-based communication is that the sensors have inadequate power supplies, while the fusion center has enough power to perform complicated computation algorithms. Hence, a sensor obtains each measurement based on an activation mechanism, and after that, it decides, through a triggering mechanism, whether to keep or transmit the measurement to the estimator. In traditional EBE, a binary triggering mechanism is used for local decision making on the sensor side, denoted by $\gamma_k$, defined as

$$\begin{cases} \gamma_k = 1 : \text{Measurement occurs, communication triggered} \\ \gamma_k = 0 : \text{No measurement occurs, no communication} \end{cases}$$

According to the above mentioned framework, the set of observations, including that of iteration $k$, is defined as $\mathbf{Z}_k = \{\gamma_0, \mathbf{z}_0, \dots, \gamma_k, \mathbf{z}_k\}$.

*Ternary Event-Triggering (TET)*

The counterpart to using traditional binary decision making is to use the triggering approach called the ternary event triggering (TET) mechanism. The TET mechanism is a deterministic and ternary scheduler which proposes three decision levels instead of the conventional binary decision making. The sensor in the ternary mechanism initially determines the discrepancy between its current and previously communicated measurements [14]. Later on, the TET method uses this difference to decide whether or not to send the data. The TET scheduling mechanism is specified as:

$$\gamma_k = \begin{cases} 0, & \text{if } |z_k - z_{\tau_k}| < \Delta_1 \\ 1, & \text{if } \Delta_1 \le |z_k - z_{\tau_k}| < \Delta_2 \\ 2, & \text{if } |z_k - z_{\tau_k}| \ge \Delta_2 \end{cases}, \tag{3}$$

where $\tau_k$ indicates the time of the previous transmission from the sensor to the estimator, and $\Delta_1$ and $\Delta_2$ are two triggering thresholds characterizing the ternary levels [14,19]. Inclusion of the ternary framework leads to a hybrid observation vector $\mathbf{Y}_k = \{\mathbf{y}_1, \mathbf{y}_2, \dots, \mathbf{y}_k\}$ where

$$\mathbf{y}_i = \begin{cases} \{z_i : z_i \in (z_{\tau_i} - \Delta_1 \,, \, z_{\tau_i} + \Delta_1)\} & \text{if } \gamma_i = 0 \\ z_i^{(Q)} \wedge \{z_i : z_i \in (z_{\tau_i} + \Delta_1 \,, \, z_{\tau_i} + \Delta_2) \vee z_i \in (z_{\tau_i} - \Delta_2 \,, \, z_{\tau_i} - \Delta_1)\} & \text{if } \gamma_i = 1 \\ z_i & \text{if } \gamma_i = 2 \end{cases} \tag{4}$$

The immediate consequence of utilizing the TET framework is the accessibility of the below hybrid measurements at the fusion center:

1. *Set-Valued Data*: When $\gamma_k = 0$, the circumstances of the ternary mechanism are not satisfied; hence the sensor does not transfer any measurement to the estimator. In this scenario, fusion center does not have the specific value of the existing sensor's observations. However, based on secondary information (i.e., the observation $z_k$ associated with the following set $(z_{\tau_k} - \Delta_1 \,, \, z_{\tau_k} + \Delta_1)$), the FC knows which observations belong to which set.

2.  *Set and Quantized-Valued Data*: When $\gamma_k = 1$, the estimator lacks the exact value of the current measurement ($z_k$), though it has access to the quantized version of the current measurement, which is transferred to the FC. Meanwhile, the FC has access to the set-valued data (i.e., the observation $z_k$ belongs to either of the following two sets $(z_{\tau_k} + \Delta_1 , z_{\tau_k} + \Delta_2)$ or $(z_{\tau_k} - \Delta_2 , z_{\tau_k} - \Delta_1)$). This additional secondary information contributes to the enhancement of the approximation efficiency through quantized measurements [14,19].

3.  *Point-Valued Data*: When $\gamma_k = 2$, the TET mechanism conveys the specific measurement of the sensor $z_k$ to the FC.

## 3. The Proposed TETSO Framework

Solving engineering problems in CPSs poses numerous challenges. Multi-objective optimization is used to determine Pareto optimal solutions for each objective when more than one objective is specified [24]. The multi-objective nature of such real-world practical problems is the main characteristic that makes their solution difficult. For a given problem that has more than one key factor objective, all of these objectives can not be satisfied simultaneously, meaning there are no unique optimal solutions.

Therefore, multi-objective optimization is the process of finding Pareto optimal solutions in the objective space, where each objective component of any Pareto optimal solution may only be enhanced by degrading at least one of its other objective components. To find solutions for multi-objective problems, one should provide a set of Pareto optimal solutions in order to make the best trade-off between the multiple underlying objectives.

In 1995, Eberhart and Kennedy invented a meta-heuristic algorithm called particle swarm optimization (PSO) for single-objective problems [25]. Figure 1 shows a general PSO algorithm (the definitions and symbols are explained later in this section). This algorithm is a stochastic optimization technique inspired by natural swarm behavior such as bird flocking and fish schooling [26,27]. This approach discovers an optimal solution by moving a large number of particles, or prospective solutions, throughout the search space while following the current best particle positions. Each particle traces the best location, referred to as the best solution, found in its path. When particle $i$ finds a location that is better than all former locations that it has found, it stores that location as a new current best solution for particle $i$, $P_{best(i)}^{(t)}$. If $P_{best(i)}^{(t)}$ is better than the current global best solution, $G_{best}$, then $G_{best}$ is updated with $P_{best(i)}^{(t)}$ [28]. Initially, $G_{best}$ and $P_{best(i)}^{(t)}$ are set to zero. During the evolution of the path of particle $i$, it considers its own best location and the best global solution that the swarm has attained so far. In the PSO each particle firstly updates its velocity by considering its current position, current velocity, $P_{best(i)}^{(t)}$, and $G_{best}$. Then, the particle adapts its position using the updated velocity at every iteration. The mathematical model of the PSO algorithm is as follows,

$$V_i^{(t+1)} = wV_i^{(t)} + c_1 r_1 \left( P_{best(i)}^{(t)} - x_i^{(t)} \right) + c_2 r_2 \left( G_{best}^{(t)} - x_i^{(t)} \right), \tag{5}$$

$$X_i^{(t+1)} = X_i^{(t)} + V_i^{(t+1)} \tag{6}$$

for $1 \leq i \leq N_{SP}$ where $t$ is the completed number of iterations and $N_{SP}$ is the number of particles. As well, $X_i^{(t)}$ is the current position of the particle $i$ at iteration $t$. The velocity of particle $i$ at iteration $t + 1$ is shown as $V_i^{(t+1)}$. Moreover, $w$ is an inertia weight which determines the rate of a particle's previous velocity to its current velocity, and the terms $c_j$ and $r_j$, $j = 1, 2$, represent acceleration coefficients and uniform random numbers, respectively, distributed between 0 and 1. The concepts determining the updates of a particle's position and velocity are demonstrated in Figure 1. The first term (i.e., $wV_i^{(t)}$), on the right hand side (RHS) of Equation (5), provides PSO with the exploration ability, whereas the 2nd and 3rd terms (i.e., $c_1 r_1 (P_{best(i)}^{(t)} - x_i^{(t)})$ and $c_2 r_2 (G_{best}^{(t)} - x_i^{(t)})$) define the particle-best calculation and collective global-best calculation of the particles, respectively. The PSO

begins by randomly positioning $N_{SP}$ particles in the search area, and while conducting the required iterations, the particles' velocities by using Equation (5) are computed. Following the particles' velocities calculation, by using Equation (6) the positions of particles can then be computed. The process of re-positioning of the particles continues until a certain completion condition is fulfilled.

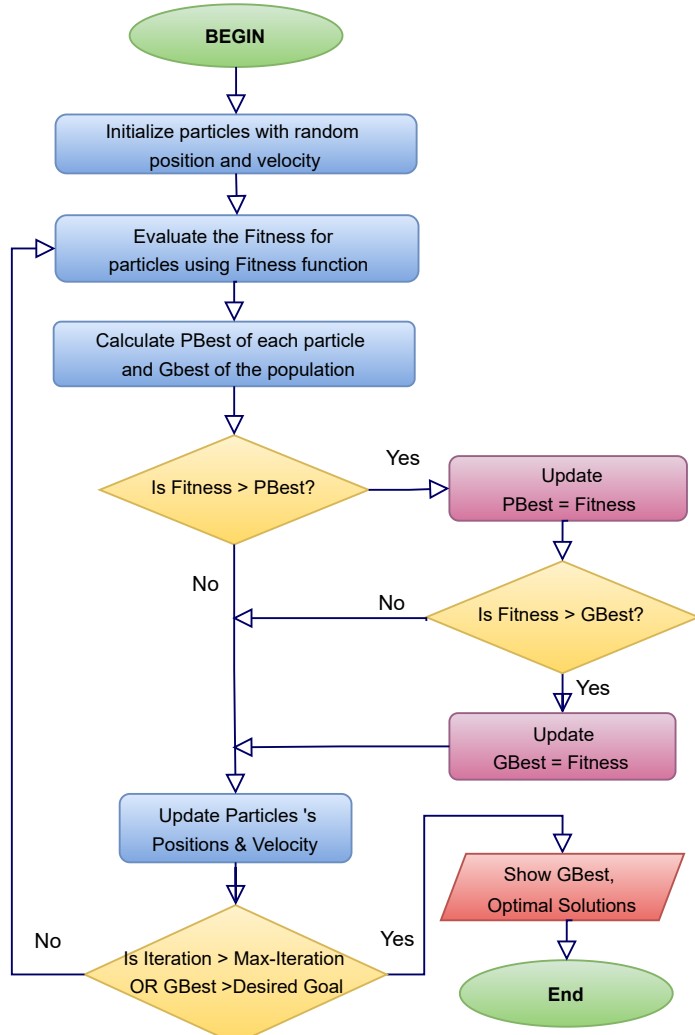

**Figure 1.** Flowchart of the general PSO algorithm.

As stated in Section 2, the TET mechanism is a ternary scheduler with three decision levels. Our proposed approach, instead of using heuristics, chooses the decision levels automatically using multi-objective particle swarm optimization and improves the selection criteria of threshold values that feed into the ternary state estimation.

In this model, an open-loop and event-based architecture is considered when a remote sensor transfers its measurement to the fusion center exclusively in case a specific event happens. The TETSO framework modifies the PSO algorithm by adding sets of thresholds representing the TET approach. The sets of thresholds are defined as follows for iteration $k$, where $\Pi_{L,k}$ is the left set of threshold values and $\Pi_{W,k}$ is the right set:

$$
\begin{aligned}
\Pi_{L,k} &= \{\Delta_{1,k}, \Delta_{2,k}, \dots, \Delta_{m,k}\} \text{ for } j = 1 \dots m, \\
\Pi_{W,k} &= K_k \cdot \Pi_{L,k} \text{ for } j = 1 \dots m
\end{aligned}
\tag{7}
$$

where $m$ is the number of thresholds in each set, $K_k$ is a coefficient whose value changes randomly with each iteration, and each $\Delta_{j,k}$ is randomly defined at each iteration within a

predefined range. The simulations section will define the specific ranges of these random values. It should be noted that the TETSO framework will look for Pareto optimal solutions that will satisfy the following constraints in the sets of thresholds, which are described as follows for $\Pi_{L,k}$ and $\Pi_{W,k}$, respectively,

$$
\begin{aligned}
\text{Constraint 1}: \quad & \Delta_{j,k} > \Delta_{j,k-1} \ \text{ for } j = 1 \ldots m \\
\text{Constraint 2}: \quad & \Delta_{j+1,k} > \Delta_{j,k} \ \text{ for } j = 1 \ldots m-1
\end{aligned}
\tag{8}
$$

Constraint 1 ensures that the values of the thresholds in $\Pi_{L,k}$ and $\Pi_{W,k}$ increase with the next iteration of $k$, and constraint 2 guarantees that the values in each set remain sorted. The aforementioned set values of the thresholds are received by the objective function of the TETSO framework as input. The framework attempts to increase the amount of quantized data communication and simultaneously decrease the point-valued and set-valued data transmission, which lowers the communication overhead and the cost [14].

### 4. Simulation Results and Discussion

This section discusses the simulation experiments developed to assess the effectiveness of the proposed architecture. The simulation parameters used in this section are based on previous work [14,29], studying observation-driven communication in WSNs, which determined that the optimal scheduling strategy parameters which allow estimate performance become comparable with full measurements under a moderate transmission rate. In particular, using the same simulation details, we compare the TET mechanism [14] with the proposed TETSO algorithm.

The following section uses the notation and development presented in the recent literature on state estimation problems [29]. The following are considered as an object's position and velocity,

$$
x_k = \begin{bmatrix} 0.8 & 1 \\ 0 & 0.95 \end{bmatrix} x_{k-1} + w_k \, .
$$

The sensor measures the position and velocity based on the observation model stated as follows,

$$
z_k = \begin{bmatrix} 0.7 & 0.6 \end{bmatrix} x_k + v_k \, .
$$

In this simulation, the represented outcomes are calculated over a Monte Carlo (MC) simulation of 100 runs with $N_{SP} = 30$ particles. The object's velocity and position in each simulation iteration change randomly to provide a fair experimental benchmark. The parameters used in this simulation are shown in the Table 1.

**Table 1.** Table of Parameters Used Throughout the Simulation.

| | |
|---|---|
| $N_{SP} = 30$ | |
| MC simulation = 100 | Max iteration = 200 |
| $c_1 = 1.4962$ | $c_2 = 1.4962$ |
| $V_1 = 0$ | $X_1 = 0$ |

This simulation compares the five following filters: (1) Full-rate estimation of the Kalman filter (KF); (2) Open-loop event-based estimation of KF; (3) Full-rate estimation of particle filter (PF); (4) Event-based estimation of particle filter with binary decision; and (5) The results of proposed TETSO architecture. KF, which is suitable for linear systems and Gaussian white noise [30], is an efficient recursive algorithm that estimates unknown variables based on measurements observed over time and is used to estimate unknown variables optimally when they are not directly measurable but indirect measurements are available [15,31]. PF is a Bayesian-based estimation algorithm based on conditional probability density, developed by Gordon et al. [32] for addressing applications that are nonlinear and non-Gaussian. Without making assumptions about the state space model or the state distributions, PF provides a mechanism for producing samples from the

appropriate distribution. The values $c_1 = c_2 = 1.4962$ have been considered for the TETSO, and the number of particles within the swarm, $N_{SP}$, is set to be 30. The multi-objective problem consists of three objectives, and the repository of previously discovered solutions is 100 in size.

The completion criterion for the particle swarm optimization is the maximum number of iterations, set to be 200. The reported results are calculated over various communication rates. The matrix consisting of a pair of sets of $\Pi_{L,k}$ and $\Pi_{W,k}$ is considered, where $\Pi_{L,k}$ includes a set of $m = 6$ numbers selected randomly at each iteration $k$ from 0.001 to 10 (i.e., $0.001 \leq \Delta_{j,k} \leq 10$), and $\Pi_{W,k}$ is determined by Equation (7) where $K_k$ is chosen randomly at each iteration from the range 1.0001 to 2. As mentioned in the previous section, constraints 1 and 2 are satisfied by the $\Delta_{j,k}$ values for the proposed TETSO.

In this experiment, we consider multi-objective optimization, which is concerned with optimization problems that involve more than one objective function to be optimized simultaneously; thus, no single optimal solution may exist. Therefore, the algorithm provides a set of Pareto optimal solutions. In multi-objective optimization problems, all the components of the Pareto optimal solution set are believed to be satisfactory designs. The suggested TETSO will identify a set of Pareto optimal solutions that are the optimal threshold values. Figure 2 illustrates the search history of the TETSO over 15,000 simulations with the points indicated by red stars being the Pareto optimal answers. Pareto optimal solutions of multi-objective PSO with Subfigures are shown in Figure 3. Each subfigure emphasizes different angles. Mean squared error (MSE) calculates the mean of the squares of between the actual values and the estimated values. The position MSE against different values of the $\Delta_{j,k}$ among the ternary levels of the TETSO is presented in Figure 4. The comparison of position MSE among different transmission rates is presented in Figure 5. The results demonstrate that TETSO shows remarkable performance in terms of reducing the $\Delta_{i,k}$ rates compared to the other filters. The results show that the suggested architecture contributes to a lower communication rate and surpasses its counterparts, which verifies that the suggested approach is effective.

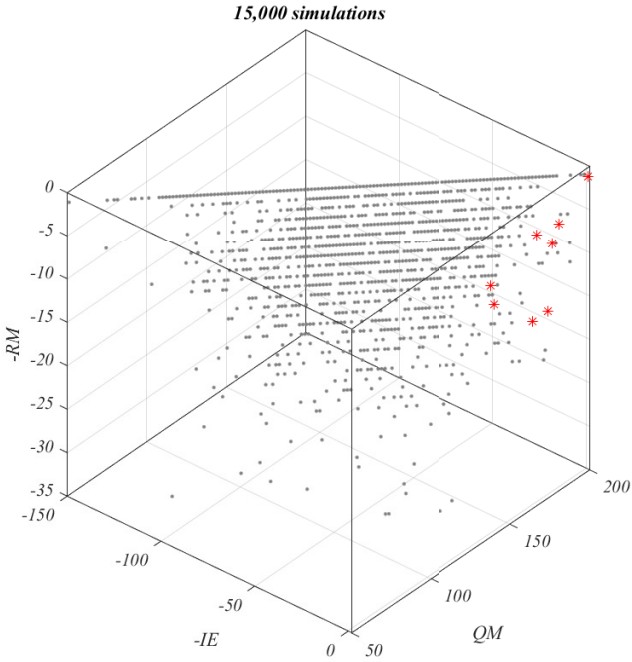

**Figure 2.** Search space history of multi-objective PSO with some marked designs. The Pareto optimal solutions are highlighted by red stars and non-dominated solutions with dark points.

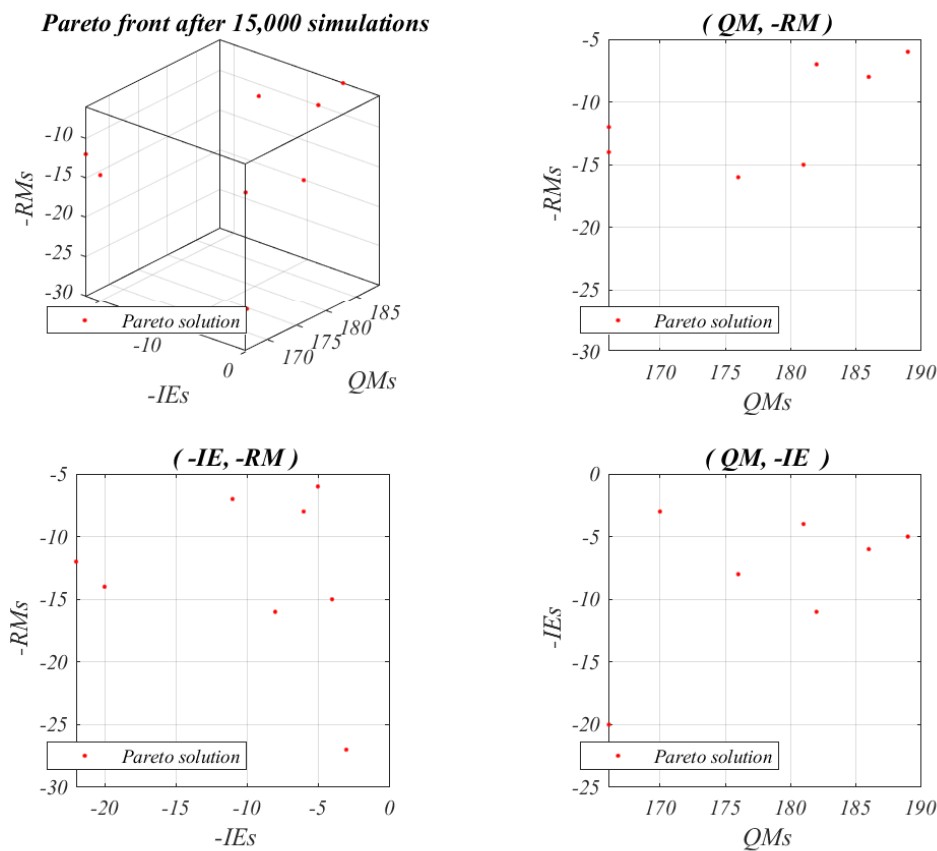

**Figure 3.** Pareto optimal solutions of multi-objective PSO with some marked designs. Each subfigure emphasizes different angles.

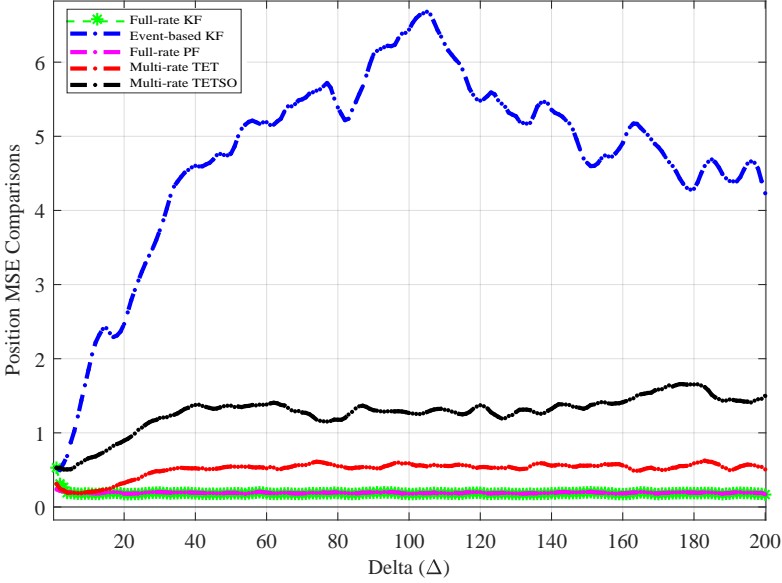

**Figure 4.** The MSE comparison across various values of the $\Delta_{j,k}$.

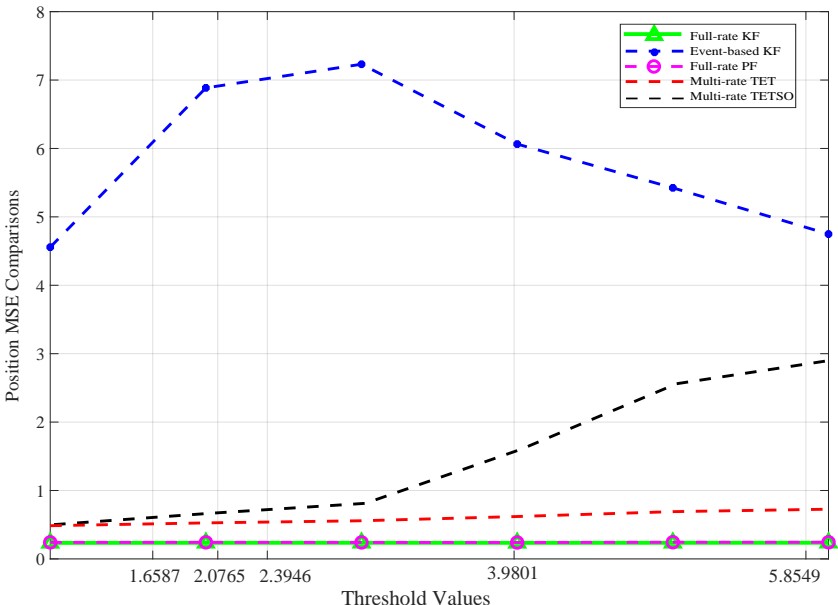

**Figure 5.** The MSE comparison across communication rates achieved by TETSO.

Table 2 demonstrates the analogy between the set-valued, quantize-valued, and point-valued measurements against the changing values of $\Delta_{j,k}$, of which the results in the eight best solutions are shown. It should be noted that the suggested framework has the potential to decrease the overall communication rate by transmitting more quantized measurements and decreasing the measurement's transmission rate in the case of the event epochs.

**Table 2.** Total number of communicated observations in terms of idle epoch (IE), transient (QE), and raw measurements (RM).

| Group | I | II | III | IV | V | VI | VII | VIII |
|-------|-----|-----|-----|-----|-----|-----|-----|------|
| IE | 6 | 3 | 20 | 4 | 5 | 22 | 8 | 11 |
| QM | 186 | 170 | 166 | 181 | 189 | 166 | 176 | 182 |
| RM | 8 | 27 | 14 | 15 | 6 | 12 | 16 | 7 |

## 5. Conclusions

The TETSO solution presented in this paper works for a wide range of applications in the domain of CPS, where the problem has multiple objectives that must be optimized to generate efficient solutions. In this article, we have presented an effective approach for the calculation of a Pareto optimal set of threshold values in the ternary event-based estate estimation. In the state estimation problems, the triggering mechanism depends on a predetermined threshold value; hence, selecting an efficient threshold value has become a significant matter. To achieve this, we proposed a multi-objective framework called ternary event triggering swarm optimization (TETSO), which effectively selected the TET mechanism's threshold values. The suggested approach yields a non-Gaussian distributed event-based approximation framework that is also systematic. The simulation results confirm that the suggested technique is capable of designing and optimizing the TET mechanism's threshold values and, furthermore, can be applied to any TET mechanism. The proposed technique relies less on the initial threshold value to start the optimization and limits human involvement in the optimization process. Finally, the proposed framework presents an effective technique of intelligently using hybrid sets of information, resulting in concurrent decrements in communication overhead as well as MSE.

**Author Contributions:** Writing—original draft preparation and investigation, software, methodology, visualization: S.D.; writing—review, methodology and supervision, T.F. All authors have read and agreed to the published version of the manuscript.

**Funding:** This research received no external funding.

**Institutional Review Board Statement:** Not applicable.

**Informed Consent Statement:** Not applicable.

**Data Availability Statement:** All codes developed in this work will be made available from the corresponding author upon reasonable request.

**Conflicts of Interest:** The authors declare no conflict of interest.

## Abbreviations

List of abbreviations used throughout this paper:

| Abbreviation | Definition |
| --- | --- |
| ADAI | Adaptive distributed artificial intelligence |
| APSP | Adaptive Particle Swarm Optimization |
| CPS | Cyber-physical system |
| EBE | Event-based estimation |
| IE | Idle epoch |
| KF | Kalman filter |
| MSE | Mean square error |
| PF | Particle filter |
| PSO | Particle swarm optimization |
| QM | Quantize measurements |
| RM | Row measurements |
| SOD | Send on delta |
| TET | Ternary event triggering |
| TETSO | Ternary event triggering swarm optimization |

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
