# Peer review of "A Novel Approach for Selecting Effective Threshold Values in Ternary State Estimation Using Particle Swarm Optimization"

_applsci, doi:10.3390/app122110693_

Round 1
Reviewer 1 Report
The document is well structured and complies with an organized methodology for the research. The results shown are satisfactory from comparisons with other methods. The authors are encouraged to perform laboratory tests as a further development after the simulations.
It is important that the authors review the papers https://ieeexplore.ieee.org/document/8623820 and https://spectrum.library.concordia.ca/id/eprint/984231/13/Davar_MASc_F2018.pdf to decrease the coincidence rate. This does not imply self-plagiarism on the part of the authors, but it is recommended that the similarity index be less than 20%.
It is recommended to expand the references with more current citations if possible.
Author Response
Please, find attached.

Reviewer 2 Report
Title: A Novel Approach for Selecting Effective Threshold Values in Ternary State Estimation using Particle Swarm Optimization
Comments: The authors have well explored the concept of PSO as an optimizer to identify Pareto optimal set values of the 4 thresholds. Some of the issues that may be addressed in the revised version are:
1. The objective of the research and the the state of art may be written point-wise in the Introduction.
2. For most of the equations, this is not clear if they are derived or cited? as they look like the basic equations.
3. The authors may include a table of notations and abbreviations for ease in readabiity.
4. some of the recent work may be included in the discussions. ex: "Low-energy PSO-based node positioning in optical wireless sensor networks"; ""ADAI and adaptive PSO-based resource allocation for wireless sensor networks"; "Verma, K. K., & Singh, B. M. (2021). Deep Multi-Model Fusion for Human Activity Recognition Using Evolutionary Algorithms. International Journal of Interactive Multimedia & Artificial Intelligence, 7(2)" etc.
5. The authors may prepare a simulation scenario table and include their assumption criteria for the respective simulation comparisons.
Author Response
Please, find attached.

Reviewer 3 Report
Dear Authors,
The ternary state estimation using PSO is your major idea. The idea of the article shall be considered under the cyber physical systems. However, the limitations of the article listed below are more crucial.
1. First, The objective of this article has not been clearly defined in section 1. The applications of cyber physical systems, the need for novel threshold value selection, importance of multi-objective cases, sensor environment, issues in computation overhead etc. are not presented neatly. The research problem (or application problem) and the contributions must be listed in section 1.
2. Section 2 shows problem formulation. On what basis the problem has been identified? This article has not provided the details of relevant techniques and limitations. On the strong scope, the problem has to be defined for showing better reading view. A detailed and tabulated comparisons are expected on existing techniques.
3. The proposed system (section 3) is not effectively explained with necessary models and technical benefits. Figure 1 shows more generic idea and it is not providing the idea of this article.
4. Basically, the article focused in to "Applied Sciences" require a worthy notes of proposed models that are justifying the contributions in particular application domain. This is completely missed and application benefits are not exposed. Similarly, multi-objective sensor attributes and sensor models are not denoted in detail.
5. Result section has limited information. There is no sign for getting application benefits, overhead control, better communication rate, simulation details, tool and test environment and valid contribution supports. Only MSE is identified.
6. References 6 and 7 are looking similar to this article. The application benefits of this new approach are not effectively compared and results are not provided in this concern. The readers expect a neatly narrated article in all aspects.
7. Conclusion shall be improved with more crucial benefits and application stands.
In general, this article has some idea and presentation points yet it has lack of technical details, field-based application benefits, and proofs. This will confuse any reader. Please improve all as mentioned in future. However, I reject this article in present form. Thanks.
Author Response
Please, find attached.

Round 2
Reviewer 3 Report
Still the results are not properly improved in order to prove application specific needs. That can be improved.
